# Understanding Federated Learning through Loss Landscape Visualizations: A Pilot Study

**Ziwei Li, Hong-You Chen, Han-Wei Shen, Wei-Lun Chao**
Department of Computer Science and Engineering
The Ohio State University
{li.5326, chen.9301, shen.94, chao.209}@osu.edu

## Abstract

Federated learning aims to train a machine learning model (e.g., a neural network) in a data-decentralized fashion. The key challenge is the potential data heterogeneity among clients. When clients' data are non-IID, federatedly learned models could hardly achieve the same performance as centralizedly learned models. In this paper, we conduct the very first, pilot study to understand the challenge of federated learning through the lens of loss landscapes. We extend the visualization methods developed to uncover the training trajectory of centralized learning to federated learning, and explore the effect of data heterogeneity on model training. Through our approach, we can clearly visualize the phenomenon of model drifting: the more the data heterogeneity is, the larger the model drifting is. We further explore how model initialization affects the loss landscape, and how clients' participation affects the model training trajectory. We expect our approach to serve as a new, qualitative way to analyze federated learning.

## 1 Introduction

Federated learning (FL) aims to train a machine learning model (e.g., a neural network) in a data-decentralized fashion [10]. Namely, data holders (i.e., clients) do not need to share their data with a centralized server for model training. This largely protects data privacy and ownership.

Training a model in an FL manner, however, is by no means trivial. As data are decentralized, model training (i.e., parameter update) must take place at the clients' local sites, followed by an aggregation process to produce a single, global model. When clients data are of different distributions (i.e., non-IID) — a common scenario in reality — federatedly learned models have been shown to perform poorly compared to centralizedly learned models. Extensive studies have thus been conducted to understand and alleviate such an issue [16, 11, 1]. However, most of

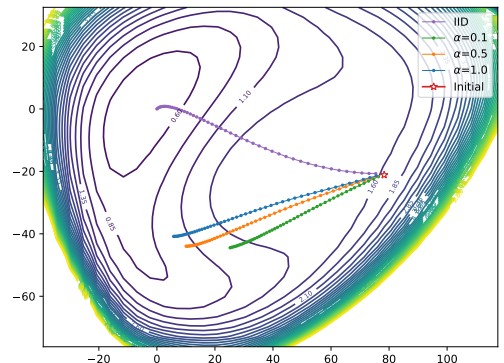

Figure 1: **Visualization of the training trajectory of FEDAVG [18].** We train a ConvNet under different non-IID conditions of CIFAR-10 [12]. Smaller $\alpha$ means a severer non-IID condition [9]. We collect the global model after each round, and plot the loss surface using the aggregated training data. As shown, the training trajectory under non-IID data gradually deviates away from that under IID data, and ends at a larger loss value.

these studies were presented in quantitative ways. There is a lack of user-friendly, qualitative ways (e.g., visualizations) to understand the inner working and challenges of federated learning.

Workshop on Federated Learning: Recent Advances and New Challenges, in Conjunction with NeurIPS 2022 (FL-NeurIPS'22). This workshop does not have official proceedings and this paper is non-archival.

In this paper, we explore the use of visualizations to understand the loss landscape and model training trajectory of federated learning. Visualizing the loss landscape has been shown as an intuitive way to understand *centralized* neural network training and how different network architectures, optimizers, and hyperparameters (e.g., batch size, learning rate) play their roles [14, 7]. Compared to centralized learning, federated learning has a more complicated training procedure as it usually involves iterations between local training (of multiple models in parallel) and global aggregation.

To perform visualizations in a meaningful way, we first discuss what to visualize in FL. We identify **two loss landscapes**, the loss function computed on each client's data or on the aggregated data, and **two training trajectories**, local model training or global models aggregated after different iterations. We show how to extend the existing approaches [14, 7] to visualize them. We then use visualizations to explore the effect of data heterogeneity on the loss landscape and training trajectory. Our goal is to understand how data heterogeneity makes the optimization in FL harder, and how existing solutions (e.g., pre-training) could alleviate it.

We obtain the following insights through our visualizations.

- Treating the trajectory on IID data as the *ideal* trajectory one can obtain in FL, we show that when the data becomes less IID, the trajectory would deviate away from the ideal trajectory, or even converge to a different (and worse) local minima. This demonstrates the phenomenon of model drifting, one major cause of performance degradation in FL [17].
- With proper model initialization, the trajectories under different non-IID degrees (including IID) would enter the same basin of the loss landscape. This provides an explanation of why pre-training could largely improve FL [3].
- Partial client participation makes the training trajectory less stable.

For future work, we will extend our visualizations to compare different local training [11, 1, 15] and model aggregation [2, 13, 20] approaches on more datasets, as well as other hyperparameter settings (e.g., local epochs). We expect that our visualizations to serve as an analytics tool for other future research in FL.

## 2 Background

We provide backgrounds of federated learning and visualizations of loss landscapes.

**Federated learning (FL).** In FL with $M$ clients, each client has a data set $\mathcal{D}_m = \{(\boldsymbol{x}_i, y_i)\}_{i=1}^{|\mathcal{D}_m|}$. The optimization problem to solve can be formulated as

$$\min_{\boldsymbol{\theta}} \ \mathcal{L}(\boldsymbol{\theta}) = \sum_{m=1}^{M} \frac{|\mathcal{D}_m|}{|\mathcal{D}|} \mathcal{L}_m(\boldsymbol{\theta}), \quad \text{where} \quad \mathcal{L}_m(\boldsymbol{\theta}) = \frac{1}{|\mathcal{D}_m|} \sum_{i} \ell(\boldsymbol{x}_i, y_i; \boldsymbol{\theta}). \tag{1}$$

Here, $\boldsymbol{\theta}$ is the model parameter; $\mathcal{D} = \cup_m \mathcal{D}_m$ is the aggregated data set from all clients; $\mathcal{L}_m(\boldsymbol{\theta})$ is the empirical risk computed from client $m$'s data; $\ell$ is a loss function applied to each data instance.

As clients' data are decentralized, Equation 1 cannot be solved directly (otherwise, it is centralized learning). A standard way to *relax* it is Federated Averaging (FEDAVG) [18], which iterates between two steps, local training and global aggregation, for multiple rounds of communication (indexed by $t$)

$$\textbf{Local:} \ \ \boldsymbol{\theta}_m^{(t)} = \arg\min_{\boldsymbol{\theta}} \mathcal{L}_m(\boldsymbol{\theta}), \text{ initialized with } \bar{\boldsymbol{\theta}}^{(t-1)}; \quad \textbf{Global:} \ \ \bar{\boldsymbol{\theta}}^{(t)} \leftarrow \sum_{m=1}^{M} \frac{|\mathcal{D}_m|}{|\mathcal{D}|} \boldsymbol{\theta}_m^{(t)}. \tag{2}$$

The local training is performed at all (or part of) the clients in parallel, usually with multiple epochs of SGD to produce the local model $\boldsymbol{\theta}_m$. The global aggregation is by taking element-wise average over model weights. Since local training is driven by clients' empirical risks, when clients' data are non-IID, $\boldsymbol{\theta}_m$ would drift away from each other, making $\bar{\boldsymbol{\theta}}$ deviate from the solution of Equation 1.

**Visualizations of loss landscapes.** Visualizing the optimization trajectory and its surrounding loss surface is a powerful technique to analyze the high-dimensional learning behavior and generalization of neural networks [5, 6, 7, 14]. Let $\mathcal{L}$ be the loss function and $\boldsymbol{\theta} \in \mathbb{R}^K$ be the parameter, to perform

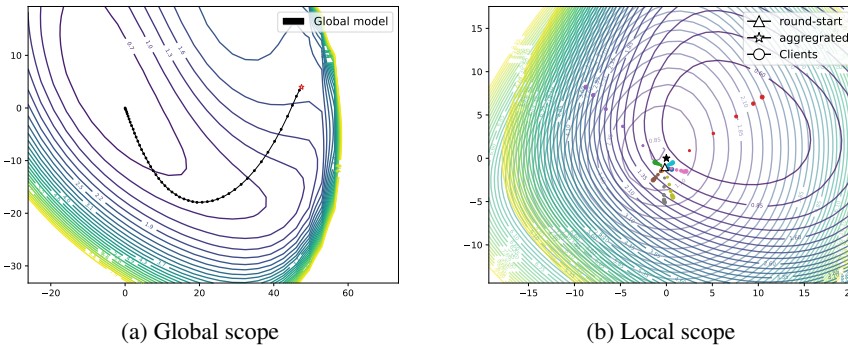

| (a) Global scope | (b) Local scope |

Figure 2: **Visualizations of the global scope and local scope.** (a) We show the trajectory of the global models over 100 rounds on the loss landscape of $\mathcal{L}$. (b) We show the trajectories from $\bar{\boldsymbol{\theta}}^{(t-1)}$ (round start $\triangle$) to $\boldsymbol{\theta}_m^{(t)}$, and to $\bar{\boldsymbol{\theta}}^{(t)}$ (aggregated $\star$), for all 10 clients at a particular round $t = 10$. On the background, we overlay two landscapes together: $\mathcal{L}$ (the minima is at the middle); $\mathcal{L}_m$ for the red client (the minima is on the right).

2D visualizations, one common way is to visualize the loss in a 2D subspace $\boldsymbol{\theta} = \boldsymbol{v}^* + \beta_1 \boldsymbol{v}_1 + \beta_2 \boldsymbol{v}_2$. Here, $\boldsymbol{v}^* \in \mathbb{R}^K$ is the center point; $\boldsymbol{v}_1 \in \mathbb{R}^K, \boldsymbol{v}_2 \in \mathbb{R}^K$ are the spanning directions; $\beta_1 \in \mathbb{R}, \beta_2 \in \mathbb{R}$ are the coefficients. That is, $(\beta_1, \beta_2)$ on the 2D visualization plane corresponds to $\mathcal{L}(\boldsymbol{\theta} = \boldsymbol{v}^* + \beta_1 \boldsymbol{v}_1 + \beta_2 \boldsymbol{v}_2)$.

To visualize the loss landscape around a specific model checkpoint, one can simply use it as $\boldsymbol{v}^*$, and randomly sample $\boldsymbol{v}_1$ and $\boldsymbol{v}_2$. To visualize the training trajectory along multiple checkpoints, one common way is to set $\boldsymbol{v}^*$ as the last checkpoint, set $\boldsymbol{v}_1$ and $\boldsymbol{v}_2$ as the top two principal components of the checkpoints, and then project each checkpoint onto the 2D subspace/plane.

# 3 Visualizations of the Loss Landscape of Federated Learning

As mentioned in section 2, FL has a more complicated training procedure than centralized learning. Specifically, FEDAVG involves multiple rounds of parallel local training (each with multiple epochs) followed by global aggregation. The goal is to learn $\bar{\boldsymbol{\theta}}$ to minimize the global loss $\mathcal{L}$ calculated on the aggregated data, while during local training, each $\boldsymbol{\theta}_m$ is trained to minimize the local loss $\mathcal{L}_m$ calculated on each client's data. To make the visualizations meaningful, we must clearly identify what to visualize, i.e., what models and on what loss functions.

To this end, we consider two scopes: global and local. At the global scope, we analyze the trajectory along the global models $\{\bar{\boldsymbol{\theta}}^{(t)}\}$ after each round, and see if it converges to the minima of $\mathcal{L}$. At the local scope, we analyze the trajectory from $\bar{\boldsymbol{\theta}}^{(t-1)}$ to $\boldsymbol{\theta}_m^{(t)}$, and to $\bar{\boldsymbol{\theta}}^{(t)}$, within each communication round. In particular, we want to see if local training makes each $\boldsymbol{\theta}_m^{(t)}$ deviate from the overall goal of minimizing $\mathcal{L}$, and if global aggregation can compensate for it. For each scope, we collect the corresponding model checkpoints, and follow the procedure in section 2 to create the visualizations.

**Experimental setup.** We conduct experiments on the CIFAR-10 [12] image classification dataset. We follow [9] to split the 50K training images into $M = 10$ portion. A coefficient $\alpha$ controls the non-IID degree. The smaller the $\alpha$ is, the larger the non-IID degree is. In the extreme cases, each client may have data from only a single class. On contrary, when $\alpha$ is large, each client is likely to have data from all the classes; the class proportion is close to uniform.

We consider two neural networks. One is the ConvNet with 6 Conv and 2 FC layers. The other is ResNet20 [8]. We apply FEDAVG [18] with 100 rounds of local training. Each round takes 5 epochs, using a learning rate of $0.01$ and a batch size 16. We consider full client participation at each round, but will investigate partial participation in section 5. We initialize the global model $\bar{\boldsymbol{\theta}}^{(0)}$ with random weights, but will investigate pre-trained weights in section 4.

**Global scope.** We show an example of the global scope in Figure 2a. We use the ConvNet and $\alpha = 0.3$ (mid non-IID), and visualize the trajectory of the global models $\{\bar{\boldsymbol{\theta}}^{(t)}\}$ on the loss landscape of $\mathcal{L}$. As shown, the global models seem to converge to a minima of $\mathcal{L}$.

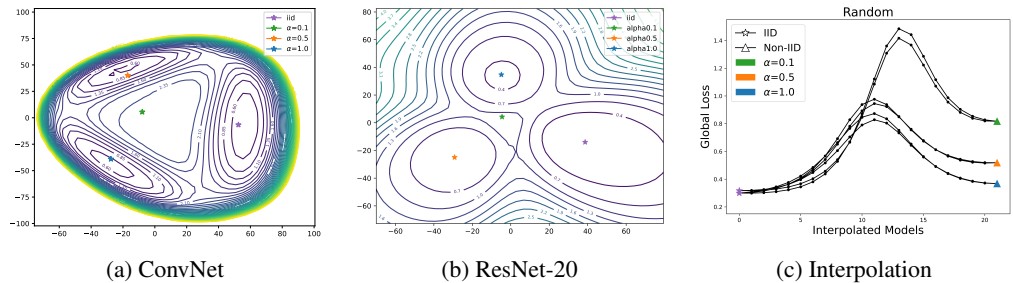

| (a) ConvNet | (b) ResNet-20 | (c) Interpolation |

Figure 3: **Visualizations of the final global models (i.e., $\bar{\theta}^{(100)}$) under the IID and different non-IID conditions, using the same random weights for initialization.** (a) Using the ConvNet. (b) Using the ResNet-20. (c) Global loss $\mathcal{L}$ along the interpolation between the IID global model and each of the non-IID global model. A higher intermediate loss indicates the existence of a larger barrier between two models.

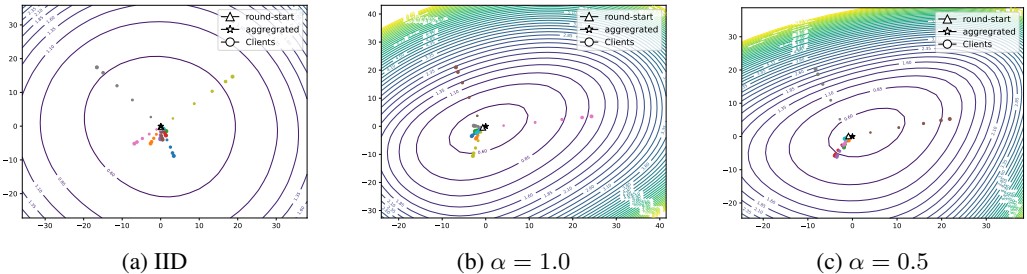

| (a) IID | (b) $\alpha = 1.0$ | (c) $\alpha = 0.5$ |

Figure 4: **Visualizations of the local scope under different non-IID conditions, at $t = 30$.**

**Local scope.** We show an example of the local scope in Figure 2b. We visualize the trajectories from $\bar{\theta}^{(t-1)}$ to $\theta_m^{(t)}$, and to $\bar{\theta}^{(t)}$, for all clients at a particular round $t = 10$. We plot two loss surfaces together, one is on $\mathcal{L}$ and the other is on $\mathcal{L}_m$ of one client. As shown, the model learned during local training moves away from the minima of the global loss $\mathcal{L}$ and towards the minima of the local loss $\mathcal{L}_m$. Nevertheless, after aggregation, $\bar{\theta}^{(t)}$ moves towards the minima of the global loss $\mathcal{L}$.

## 4 Visualizations of the Effect of Data Heterogeneity and Model Initialization

In this section, we use visualizations to explore the effect of data heterogeneity on federated learning. We consider IID, $\alpha = 1.0$ (slight non-IID), $\alpha = 0.5$, and $\alpha = 0.1$ (severe non-IID).

**Initialization with random weights.** We first visualize the final global model $\bar{\theta}^{(100)}$ and surrounding loss landscape of $\mathcal{L}$. We use the same random weights to initialize FEDAVG for the IID and different non-IID conditions, and plot the four final global models together on the landscape of global loss $\mathcal{L}$. Figure 3a shows the visualizations using the ConvNet. We find that even with the same initialization, global models under different non-IID conditions soon diverge into different loss basins. The smaller the $\alpha$ is (severer non-IID), the larger the loss is, which seems to explain the degradation of FL under non-IID conditions. A similar observation is found for ResNet-20 in Figure 3b.

To further verify our observations, we linearly interpolate between the IID and non-IID final global models, and plot the intermediate model's global loss in Figure 3c. We see a notable barrier between the IID model and the non-IID model, suggesting that they indeed enter different loss basins. To make sure that this does not happen by chance, we train another global models under the IID condition but with a different data split. (The same random weights are used for initialization.) We then perform the interpolation again. We have a quite consistent observation in Figure 3c: we can clearly see three pairs of highly consistent curves (each pair is from the two IID models to one non-IID model); all curves have large losses in the middle.

We further explore how data heterogeneity affects the local scope, as shown in Figure 4. We found that the smaller the $\alpha$ is (severer non-IID), the denser the contour is, meaning that the locally trained models quickly move away from the minima of the global loss. Moreover, for less non-IID conditions, the aggregated global model (either $\bar{\theta}^{(t-1)}$ or $\bar{\theta}^{(t)}$ at $t = 30$) is closer to the center of the basin; i.e., it is more likely to attain smaller global losses.

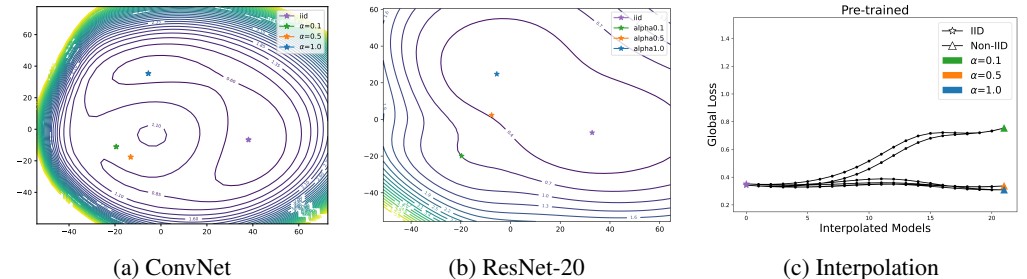

(a) ConvNet                    (b) ResNet-20                    (c) Interpolation

Figure 5: **Visualizations of the final global models (i.e., $\bar{\theta}^{(100)}$) under the IID and different non-IID conditions, using the same pre-trained weights for initialization.** (a) Using the ConvNet. (b) Using the ResNet-20. (c) Global loss $\mathcal{L}$ along the interpolation between the IID global model and each of the non-IID global model. A higher intermediate loss indicates the existence of a larger barrier between two models.

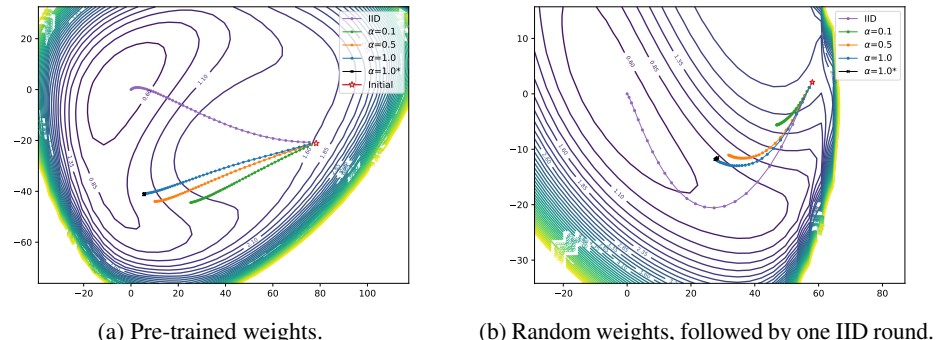

(a) Pre-trained weights.                    (b) Random weights, followed by one IID round.

Figure 6: **Visualization of the global scope, under different non-IID conditions.** We use the ConvNet. (a) We initialize FEDAVG with the ImageNet pre-trained weights. (b) We initialize FEDAVG with the random weights after one round of IID training. For each figure, we also consider running FEDAVG for more rounds, using a reduced learning rate ($\alpha = 1.0^{\star}$).

**Initialization with pre-trained weights.** Several recent work [3, 19] has empirically and quantitatively showed that pre-training significantly improves FEDAVG. It closes the performance gap between centralized learning and federated learning under non-IID conditions. Here, we want to investigate how model initialization with pre-trained weights affects FL, through visualizations. We follow [3] to pre-train the model using ImageNet data [4] that are down-sampled to the size of CIFAR-10 images.

Figure 5 shows the visualizations under the same setting as Figure 3, except the initialization. Unlike random weights, initialization with a well pre-trained model leads to a much smoother loss landscape. All the final global models (i.e., $\bar{\theta}^{(100)}$ under IID and different non-IID conditions) enter the same loss basin. The non-IID model with $\alpha = 1.0$ can even arrive at the same loss level of the IID model. Besides, we see a small barrier along the interpolation. All of these together seem to well explain why pre-training can largely improve FL, especially under non-IID conditions.

**Detailed visualizations on training trajectories.** The observation that FEDAVG initialized with pre-trained weights would enter the same loss basin under different non-IID conditions allows us to further analyze the phenomenon of model drifting (see section 2). Treating the trajectory of global models under the IID data as the *ideal* trajectory one could obtain in FL, we aim to visualize how the trajectory changes when the data becomes non-IID.

To this end, we collect all the intermediate global models $\bar{\theta}^{(t)}$ for the IID and non-IID conditions, and visualize them on the loss landscape of $\mathcal{L}$. Figure 6a shows the visualization. When the data become non-IID (i.e., from IID to $\alpha = 1.0$, $\alpha = 0.5$, and then to $\alpha = 0.1$), the trajectory gradually deviates away from the ideal trajectory (purple one, under IID data) and ends at a higher loss value.

Importantly, even if we run more rounds of FEDAVG and use a reduced learning rate, FEDAVG under non-IID data (e.g., at $\alpha = 1.0$) cannot converge to the final global model obtained under the IID data. This is shown by the black segment (i.e., $\alpha = 1.0^{\star}$) in Figure 6a. This observation may seem to conflict the observation in Figure 5c. *Namely, with pre-training, if there is no barrier between the final global models obtained under IID and non-IID data, why cannot the latter eventually converge to the*

*former after more rounds of* FEDAVG*?* To answer this question, we revisit the training procedure of FEDAVG in section 2. We note that while the overall goal of FEDAVG is to minimize $\mathcal{L}$, it does not have a direct access to $\mathcal{L}$. That is, even there is no barrier (on the landscape of $\mathcal{L}$), FEDAVG cannot sufficiently guide the final global model towards the minima of $\mathcal{L}$.

The above results demonstrate the phenomenon of model drifting, one major cause of degradation in FL. Moreover, while pre-training largely improves FEDAVG, it still cannot eliminate model drifting.

**Visualizations of the training trajectories without pre-training.** In Figure 5c, we use pre-trained weights to initialize FEDAVG. This is because with random initialization, FEDAVG will rapidly diverge into different loss basins under different non-IID conditions, making it hard to visualize the gradual deviation of the training trajectories (cf. Figure 5c). Here, we investigate if we can still visualize the phenomenon without a pre-trained model for initialization. After all, for some applications beyond natural images or natural language, pre-training might not be feasible.

We consider an analytics strategy, which is to run FEDAVG under the IID data for one round, using random weights for initialization. After that, we then run FEDAVG for different non-IID conditions, using the aggregated global model $\bar{\theta}^{(1)}$ under the IID data as the initialization. Figure 6b shows the visualization, using the same setting as in Figure 6a except for the initialization. We have a similar observation as in Figure 6a: the training trajectory under non-IID data deviates away from that under IID data; the degree of deviation is governed by the non-IID degree (i.e., $\alpha$).

## 5    Additional Analyses

We explore how other factors in federated learning, such as partial client participation, affect the loss landscape and training trajectory.

**Partial client participation.** So far, we assume that all the clients participate in each round of local training and global aggregation. In reality, clients may only participate in part of the FL process, e.g., due to communication bandwidth or quality. In Figure 7, we visualize such a case, where we randomly sample a portion of the $M = 50$ clients at each round to participate in local training and global aggregation. We conduct the experiments with both IID and non-IID data ($\alpha = 0.5$). Figure 7a shows the three trajectories using different client sampling rates (i.e., $0.2, 0.5, 0.8$). When the clients' data are IID, we found that even with different sampling rates (and fewer clients participating in each round), the three trajectories still fall into the same loss basin. Whereas under the non-IID condition, if we only use a small portion of clients at each round, the overall performance of FEDAVG could drop significantly, as shown in Figure 7b. Specifically, the purple trajectory with 20% client participation largely deviates away from the other two trajectories and finally arrives at a larger loss value.

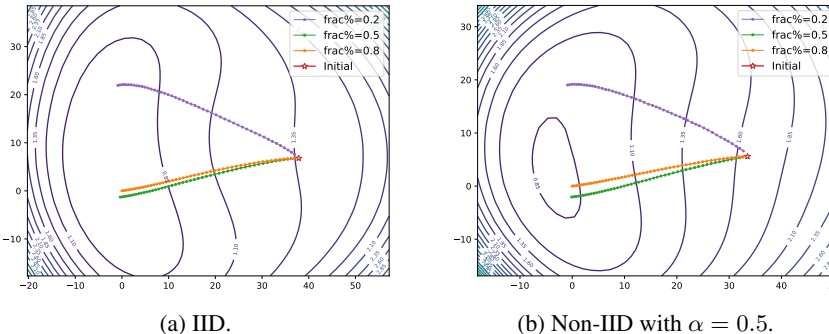

(a) IID.                                        (b) Non-IID with $\alpha = 0.5$.

Figure 7: **Visualizations on the global scope, under different levels of client participation.** We use the ConvNet. At each round, we randomly sample a portion of the $M = 50$ clients to participate in local training and global aggregation.

## 6    Conclusion

We present the very first, pilot study on using visualizations to understand federated learning. Our study clearly shows how data heterogeneity and model pre-training affect the loss landscape and

training trajectory of federated learning. We expect that our visualizations to serve as a user-friendly analytics tool for future research in federated learning. Moving forward, we plan to extend our study to different local training and model aggregation approaches on more datasets, as well as other hyperparameter settings (e.g., local epochs).

## Acknowledgments

This research is supported in part by grants from the National Science Foundation (IIS-2107077, OAC-2118240, and OAC-2112606) and Cisco Systems, Inc. We are thankful for the generous support of the computational resources by the Ohio Supercomputer Center and AWS Cloud Credits for Research.

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
