# OpenReview forum: "Understanding Federated Learning through Loss Landscape Visualizations: A Pilot Study"
_NeurIPS.cc/2022/Workshop/Federated_Learning — FL-NeurIPS 2022 Poster_

### Official Review · Reviewer_TdZ1 · 2022-10-11
**Visualizing loss landscapes in FL**

The paper conducts a qualitative study of the convergence properties of FL in various regimes by visualizing 2D projections of loss landscapes. The main regimes used are IID vs. non-IID data, pre-training vs. random initialization, and all clients vs. client sampling.

Quality:
There are numerous typos and grammatical problems throughout the paper.

Clarity:
The paper's objective is simple and clear.

Originality:
To my knowledge there have not previously been extensive visualizations of the loss landscapes in FL for non-IID settings.

Significance:
The results of this paper are as one would expect, with little novel information gained. The visualizations confirm results that have already been established quantitatively. For instance, visualizing that non-IID data disributions across clients make convergence more difficult does not lead to any novel conclusions.

Pros:
 - Visualization can help provide intuition in complicated scenarios.
 - The conclusions drawn are consistent with prior quantitative work.
 - The regimes studied are the most important regimes in FL.

Cons:
 - The intuitions shown in this paper via visualization are already well known in the community.
 - It is unclear if we can extrapolate observations made from visualizations, and it is difficult to know when intuition can be relied on vs. when it will fail.
 - Using random 2D projections of the loss landscape may lead to conclusions that do not hold in the full dimensional space.
 - Linear interpolation between models to determine the existence of a barrier neglects the non-linear, non-convex shape of the loss function. Some form of geodesic path that minimizes the weighted loss along the path would be a better measure.

I recommend that the authors supplement their qualitative conclusions with more quantitative conclusions, i.e what are the final accuracy and loss metrics obtained by the variants depicted in each plot.

---

### Official Review · Reviewer_f4kY · 2022-10-16
**A pilot study on how statistical data heterogeneity and model pre-training in FL can impact loss landscape and the training trajectory.**


1. Interesting paper. The authors could study the loss landscape when the neural is trained on other popular datasets (such as CIFAR-100, FMNIST, and MiniImageNet) as well as different architectures.
2. More details on the experimental section should be provided in the appendix.
3. Other aspects including the effect of the local epoch, batch size, training rounds, total number of clients, etc. could be studied which makes the paper more comprehensive.

---

### Official Review · Reviewer_mEQn · 2022-10-17
**Interesting research that can benefit from more realistic settings**

This paper visualizes the loss landscape and training trajectory of federated learning. Experiments on CIFAR-10 showing the 2D counter of loss landscape, projected trajectory for FedAvg under different non-IID partitioning, and local updates on different clients.

It is interesting to see such a study. I have a few suggestions,
(1) The major concern is that the experiments are only on CIFAR-10, which is a small dataset with artificial non-IID created by Dirichlet sampling of labels. I would encourage authors to consider realistic dataset with natural heterogeneity.
(2) The insights that heterogeneity causes different final solutions have been extensively studied in the literature. I would be interested to know if there are new insights from visualization?
(3) As visualization is an approximation, I am not sure if the trajectory is caused by heterogeneity, or other factors. For example, the IID curve is far from the non-IID curves, but I suspect we could get similar observation if we vary other hyperparameters like learning rate, number of local epochs etc.

---

### Public Comment · ~Debora_Caldarola1 · 2022-12-12
**On previous works studying FL through the lens of the loss landscape**

Dear authors,

Congratulations upon the acceptance of your interesting work and thank you for your contribution.
For future research, I would like to provide you with a few missing references from previous works that studied Federated Learning through the lens of the loss landscape:
- Caldarola, Debora, et al. "Improving generalization in Federated Learning by Seeking Flat Minima" (ECCV 2022). In this work, the authors analyze both homogeneous and heterogeneous FL by looking at (and comparing) convergence points and sharpness of the solutions as a proxy for generalization performances. Experiments are extended to a multitude of vision tasks (e.g. large-scale image classification, semantic segmentation) and architectures (e.g. ConvNet, ResNet, MobileNet).
- Qu, Zhe, et al. "Generalized Federated Learning via Sharpness Aware Minimization" (ICML 2022). Concurrent work to the previously mentioned ECCV22, with theoretical analyses and guarantees.
- Mendieta, Matias, et al.  "Local Learning Matters: Rethinking Data Heterogeneity in Federated Learning" (CVPR 2022), where the authors make use of landscape visualizations for analyzing the smoothness of the solution and propose a new method - FedAlign - to improve generalization in heterogeneous FL.

I hope these references will be useful for future studies in this area.

Best regards.

---

> ### Public Comment · ~Debora_Caldarola1 · 2022-12-14
> **Details on previously studied analyses**
>
> We would also like to point out the visualizations introduced in Fig. 3 and 4 of your paper reflect some of the insights and analyses proposed in [1]:
> - Fig. 2a, 2b and 14 in [1] show and compare the solutions obtained at the end of trainings carried out with three different degrees of heterogeneity on CIFAR-100 in the training and test loss landscapes. A similar analysis is proposed in Fig. 3a and 3b in this paper, differing only on the dataset (CIFAR-10) and the ResNet-20 landscape.
> - Fig. 2c, 2d, 12 and 13 in [1] show the position in the loss landscape of the local updates of three distinct clients when trained in heterogeneous and homogeneous federated scenarios, similar to the comparison proposed in Fig. 4 here (with the addition of the optimization trajectory here).
>
> Thank you for taking our feedback into account, we would be happy to provide more detailed one if you would like.
>
>
> [1] Caldarola, Debora, Barbara Caputo, and Marco Ciccone. "Improving generalization in federated learning by seeking flat minima." European Conference on Computer Vision. Springer, Cham, 2022.

---

### Decision · Program_Chairs · 2022-10-20

Accept (Poster)